# $k$-Median Clustering via Metric Embedding: Towards Better Initialization with Privacy

## Abstract

In clustering algorithms, the choice of initial centers is crucial for the quality of the learned clusters. We propose a new initialization scheme for the $k$-median problem in the general metric space (e.g., discrete space induced by graphs), based on the construction of metric embedding tree structure of the data. From the tree, we propose a novel and efficient search algorithm for good initial centers that can be used subsequently for the local search algorithm. Our method, named the HST initialization, can also be easily extended to the setting of differential privacy (DP) to generate private initial centers. Theoretically, the initial centers from HST initialization can achieve lower error than those from another popular initialization method, $k$-median++, in the non-DP setting. Moreover, with privacy constraint, we show that the error of applying DP local search followed by our private HST initialization improves previous results, and approaches the known lower bound within a small factor. Empirically, experiments are conducted to demonstrate the effectiveness of our methods.

## 1 Introduction

Clustering is an important problem in unsupervised learning that has been widely used in social network analysis, sensor networks, etc. (Punj & Stewart, 1983; Dhillon & Modha, 2001; Banerjee et al., 2005; Abbasi & Younis, 2007). The goal of clustering is to partition a set of data points into clusters such that items in the same cluster are expected to be similar, while items in different clusters should be different. This is concretely measured by the sum of distances (or squared distances) between each point to its nearest cluster center. One conventional notion to evaluate a clustering algorithms is: with high probability,

$$cost(C, D) \leq \gamma OPT_k(D) + \xi,$$

where $C$ is the centers output by the algorithm and $cost(C, D)$ is some cost function defined for $C$ on the dataset $D$. $OPT_k(D)$ is the cost of optimal (oracle) $k$-median solution on $D$. When everything is clear from context, we will use $OPT$ for short. Here, $\gamma$ is called *multiplicative error* and $\xi$ is called *additive error*. Alternatively, we may also use the notion of expected cost.

Two popularly studied clustering problems are 1) the $k$-median problem, and 2) the $k$-means problem. The origin of $k$-median dates back to the 1970's (e.g., Kaufman et al. (1977)), where one tries to find the best location of facilities that minimizes the cost measured by the distance between clients and facilities. Formally, given a set of points $D$ and a distance measure, the goal is to find $k$ center points minimizing the sum of absolute distances of each sample point to its nearest center. In $k$-means, the objective is to minimize the sum of squared distances instead. There are two popular clustering algorithms. One heuristic is the Lloyd's algorithm (Lloyd, 1982), which is built upon an iterative distortion minimization approach. In most cases, this method can only be applied to numerical data, typically in the Euclidean space. Clustering in general metric spaces (discrete spaces) is also important and useful when dealing with, for example, the graph data, where Lloyd's method is no longer applicable. A more broadly applicable approach, the local search method (Kanungo et al., 2002; Arya et al., 2004), has also been widely studied. It iteratively finds the optimal swap between the center set and non-center data points to keep lowering the cost. Local search can achieve a constant approximation ratio ($\gamma = O(1)$) to the optimal solution for $k$-median (Arya et al., 2004).

**Initialization of cluster centers.** It is well-known that the performance of clustering can be highly sensitive to initialization. If clustering starts with good initial centers (i.e., with small approximation error), the algorithm may use fewer iterations to find a better solution. The $k$-median++

algorithm (Arthur & Vassilvitskii, 2007) iteratively selects $k$ data points as initial centers, favoring distant points in a probabilistic way (see Appendix A for more details). Intuitively, the initial centers tend to be well spread over the data points (i.e., over different clusters). The produced initial center is proved to have $O(\log k)$ multiplicative error. Followup works of $k$-means++ further improved its efficiency and scalability, e.g., Bahmani et al. (2012); Bachem et al. (2016); Lattanzi & Sohler (2019). In this work, we propose a new initialization framework based on metric embedding methods, with comparable approximation error and running time as $k$-median++. Importantly, our initialization scheme can be conveniently combined with the notion of differential privacy.

**Clustering with Differential Privacy.** The concept of differential privacy (Dwork, 2006; McSherry & Talwar, 2007) has been popular to rigorously define and resolve the problem of keeping useful information for model learning, while protecting privacy for each individual. Private $k$-means problem has been widely studied, e.g., Feldman et al. (2009); Nock et al. (2016); Feldman et al. (2017), mostly in the Euclidean space. The paper (Balcan et al., 2017) considered identifying a good candidate set (in a private manner) of centers before applying private local search, which yields $O(\log^3 n)$ multiplicative error and $O((k^2 + d)\log^5 n)$ additive error. Later on, the Euclidean $k$-means errors are further improved to $\gamma = O(1)$ and $\xi = O(k^{1.01} \cdot d^{0.51} + k^{1.5})$ by Stemmer & Kaplan (2018), with more advanced candidate set selection. Huang & Liu (2018) gave an optimal algorithm in terms of minimizing Wasserstein distance under some data separability condition.

For private $k$-median clustering, Feldman et al. (2009) considered the problem in high dimensional Euclidean space. However, it is rather difficult to extend their analysis to more general metrics in discrete spaces (e.g., on graphs). The strategy of Balcan et al. (2017) to form a candidate center set could as well be adopted to $k$-median, which leads to $O(\log^{3/2} n)$ multiplicative error and $O((k^2 + d)\log^3 n)$ additive error in high dimensional Euclidean space. Gupta et al. (2010) proposed a private method for the classical local search heuristic in discrete space, which applies to both $k$-medians and $k$-means. In order to cast privacy on each swapping step, the authors applied the exponential mechanism of McSherry & Talwar (2007). Their method produced an $\epsilon$-differentially private solution with cost $6OPT + O(\triangle k^2 \log^2 n/\epsilon)$, where $\triangle$ is the diameter of point set.

**The main contributions** of this work include :

- We introduce Hierarchically Well-Separated Tree (HST), a metric embedding tree structure, to the $k$-median clustering problem. Once the HST is constructed using the data samples, we provide an efficient sampling strategy to select the initial center set, with an approximation factor $O(\log\min\{k, \triangle\})$ in the non-private setting, which is $O(\log\min\{k, d\})$ when $\triangle = O(d)$ (e.g., bounded data). This improves the $O(\log k)$ error of $k$-means/median++ in e.g., the lower dimensional Euclidean space.

- The main strength of our HST initialization method is that it could be effectively adapted to differential privacy (DP). The so-called DP-HST algorithm is $\epsilon$-DP and outputs initial centers with $O(\log n)$ multiplicative error and $O(\epsilon^{-1}\triangle k^2 \log^2 n)$ additive error. Moreover, running DP-local search starting from this initialization gives $O(1)$ multiplicative error and $O(\epsilon^{-1}\triangle k^2 (\log\log n)\log n)$ additive error, which improves previous results towards the well-known lower bound $O(\epsilon^{-1}\triangle k \log(n/k))$ on the additive error (Gupta et al., 2010) within a small $O(k\log\log n)$ factor. To our knowledge, this is the first center initialization method with differential privacy guarantee and improved error rate in general metrics.

- We conduct experiments on simulated and real-world datasets to demonstrate the effectiveness of our methods. In both non-private and private settings, our proposed HST-based approach achieves smaller cost at initialization than $k$-median++, which may also lead to improvements in the final clustering quality. Our private algorithm can also save computational costs by reaching a good solution with fewer iterations.

## 2 PRELIMINARIES

**Definition 2.1** (Differential Privacy (DP) (Dwork, 2006))**.** *If for any two adjacent data sets $D$ and $D'$ with symmetric difference of size one, for any $O \subset Range(\mathbb{A})$, an algorithm $\mathbb{A}$ satisfies*

$$Pr[\mathbb{A}(D) \in O] \leq e^\epsilon Pr[\mathbb{A}(D') \in O],$$

*then algorithm $\mathbb{A}$ is said to be $\epsilon$-differentially private.*

Intuitively, differential privacy requires that after removing any observation, the output of $D'$ should not be too different from that of the original dataset $D$. Smaller $\epsilon$ indicates stronger privacy, which, however, usually sacrifices utility. Thus, one of the central topics in differential privacy literature is to balance the utility-privacy trade-off.

To achieve DP, it is common to introduce noise to the data or the algorithm output. The *Laplace mechanism* adds Laplace($\eta(f)/\epsilon$) noise to the output, which is known to achieve $\epsilon$-DP. The *exponential mechanism* is also a tool for many DP algorithms. Let $O$ be the set of feasible outputs. The utility function $q : D \times O \to \mathbb{R}$ is what we aim to maximize. The exponential mechanism outputs an element $o \in O$ with probability $P[\mathbb{A}(D) = o] \propto \exp(\frac{\epsilon q(D,o)}{2\eta(q)})$, where $D$ is the input dataset and $\eta(f) = \sup_{|D-D'|=1} |f(D) - f(D')|$ is the sensitivity of $f$. Both mechanisms will be used in the design of our proposed DP approach.

$k$-**Median Clustering.** Following the framework of Arya et al. (2004); Gupta et al. (2010), the problem of $k$-median clustering (DP and non-DP) studied in our paper is stated as below.

**Definition 2.2** ($k$-median). *Given a universe point set $U$ and a metric $\rho : U \times U \to \mathbb{R}$, let $D \subseteq U$ be a set of demand points. The goal of DP $k$-median is to pick $F \subseteq U$ with $|F| = k$ to minimize*

$$\textit{Private } k\textit{-median:} \quad cost(F, D) = \sum_{v \in D} \min_{f \in F} \rho(v, f). \tag{1}$$

*At the same time, the output $F$ is required to be $\epsilon$-differentially private to $D$. For standard non-private $k$-median, we assume $U = D$ and minimize*

$$k\textit{-median:} \quad cost(F, U) = \sum_{v \in U} \min_{f \in F} \rho(v, f). \tag{2}$$

*We may drop "$F$" and use "$cost(D)$" or "$cost(U)$" if there is no risk of ambiguity.*

To better understand the DP clustering problem, we provide an real-world example as follows.

**Example 2.1.** *Consider $U$ to be the universe of all users in a social network (e.g., Twitter). Each user (account) is public, but also has some private information that can only be seen by the data holder. Let $D$ be users grouped by some information that might be set as private. Suppose a third party plans to collaborate with the most influential users in $D$ for e.g., commercial purposes, thus requesting the cluster centers of $D$. In this case, we need a strategy to safely release the centers, while protecting the individuals in $D$ from being identified (since the membership of $D$ is private).*

The local search procedure for $k$-median proposed by Arya et al. (2004) is summarized in Algorithm 1. First we randomly pick $k$ points in $U$ as the initial centers. In each iteration, we search over all $x \in F$ and $y \in U$, and do the swap $F \leftarrow F - \{x\} + \{y\}$ such that $F - \{x\} + \{y\}$ improves the cost of $F$ the most (if more than factor $(1 - \alpha/k)$ where $\alpha > 0$ is a hyper-parameter). We repeat the procedure until no such swap exists. Arya et al. (2004) showed that the output centers $F$ achieves 5 approximation error to the optimal solution, i.e., $cost(F) \leq 5OPT$.

---

**Algorithm 1:** Local search for $k$-median clustering (Arya et al., 2004)

---

**Input:** Data points $U$, parameter $k$, constant $\alpha$
**Initialization:** Randomly select $k$ points from $U$ as initial center set $F$
**while** $\exists\, x \in F, y \in U$ *s.t.* $cost(F - \{x\} + \{y\}) \leq (1 - \alpha/k)cost(F)$ **do**
    Select $(x, y) \in F_i \times (D \setminus F_i)$ with $\arg\min_{x,y}\{cost(F - \{x\} + \{y\})\}$
    Swap operation: $F \leftarrow F - \{x\} + \{y\}$
**Output:** Center set $F$

---

## 3   Initialization via Hierarchically Well-Separated Tree (HST)

In this section, we propose our new initialization scheme for $k$-median clustering, and provide our analysis in the non-private case solving (2). The idea is based on the metric embedding theory. We will start with an introduction to the main tool used in our approach.

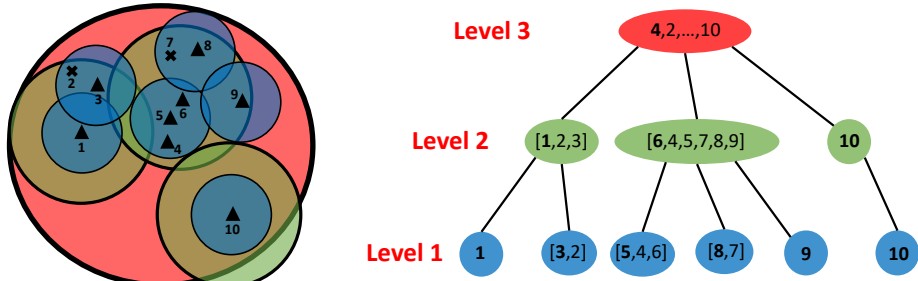

Figure 1: An illustrative example of a 3-level padded decomposition and corresponding 2-HST. **Left:** The thickness of the ball represents the level. The color corresponds to the levels in the HST in the right panel. "△"'s are the center nodes of partitions (balls), and "×"'s are non-center data points. **Right:** The resulting 2-HST generated from the padded decomposition.

## 3.1 HIERARCHICALLY WELL-SEPARATED TREE (HST)

In this paper, for an $L$-level tree, we will count levels in descending order down the tree. That is, the root is level $L$, and its children form level $L - 1$, and etc. We use $h_v$ to denote the level of $v$, and $n_i$ be the number of nodes at level $i$. The Hierarchically Well-Separated Tree (HST) is based on the padded decompositions of a general metric space in a hierarchical manner (Fakcharoenphol et al., 2004). Let $(U, \rho)$ be a metric space with $|U| = n$, and we will refer to this metric space without specific clarification. A $\beta$–padded decomposition of $U$ is a probabilistic distribution of partitions of $U$ such that the diameter of each cluster $U_i \in U$ is at most $\beta$, i.e., $\rho(u, v) \leq \beta$, $\forall u, v \in U_i$, $i = 1, ..., k$. Based on the this, we will adopt the standard 2-HST for our algorithm design.

**Definition 3.1.** *Assume the smallest distance between $u, v \in U$ is 1, and $\triangle = \max_{u,v \in U} \rho(u, v)$. A 2-Hierarchically Well-Separated Tree (2-HST) is an edge-weighted rooted tree $T$, such that an edge between any pair of two nodes of level $i - 1$ and level $i$ has length at most $\triangle/2^{L-i}$.*

We briefly describe the construction of 2-HST, and place the detailed algorithm in Algorithm 7 in the Appendix A. We first find a padded decomposition $P_L = \{P_{L,1}, ..., P_{L,n_L}\}$ of $U$ with parameter $\beta = \triangle/2$. The center of each partition in $P_{L,j}$ serves as a root node in level $L$. Then, we re-do a padded decomposition for each partition $P_{L,j}$, to find sub-partitions with diameter $\beta = \triangle/4$, and set the corresponding centers as the nodes in level $L - 1$, and so on. Each partition at level $i$ is obtained with $\beta = \triangle/2^{L-i}$. This process proceeds until a node has a single point, or a pre-specified tree depth is reached. In Figure 1, we provide an example of $L = 3$-level 2-HST (left panel), along with its underlying padded decompositions (right panel).

HST induces a tree metric, which will be related to the general metric in our analysis. It is worth mentioning that, there are polynomial time algorithms for computing an *exact* k-median solution in the tree metric (Tamir (1996); Shah (2003)). However, the dynamic programming algorithms have high complexity ($O(kn^2)$ and $O(k^2nh)$ where $h$ is at least $O(\log n)$, respectively), making them unsuitable to serve the purpose of fast initialization. Moreover, it is unknown how to apply them effectively to the private case. Hence, next, we propose a novel tree search algorithm that is very efficient, produces a $O(1)$ approximation error in the tree metric, and can be extended to DP easily.

## 3.2 HST INITIALIZATION ALGORITHM

Let $L = \log \Delta$ and suppose $T$ is a level-$L$ 2-HST in $(U, \rho)$, where for simplicity we assume $L$ is an integer. For a node $v$ at level $i$, we use $T(v)$ to denote the subtree rooted at $v$. Let $N_v = |T(v)|$ be the number of data points in $T(v)$. The sampling strategy for initial centers is presented in Algorithm 2, which has two phases: subtree search and leaf search.

**Subtree search.** The first step is to identify the subtrees that contain the $k$ centers. To begin with, $k$ initial centers $C_1$ are picked from $T$ who have the largest $score(v) = N(v) \cdot 2^{h_v}$. This is intuitive, since to get a good clustering, we typically want the ball surrounding each center to include more data points. Next, we do a screening over $C_1$: if there is any ancestor-descendant pair of nodes, we remove the ancestor from $C_1$. If the current size of $C_1$ is smaller than $k$, we repeat the process until $k$ centers are chosen (we do not re-select nodes in $C_1$ and their ancestors). This way, $C_1$ contains $k$

---

**Algorithm 2:** HST initialization - NDP (Non-Differentially Private)

---

**Input:** $U, \triangle, k$
**Initialization:** $L = \log \triangle, C_0 = \emptyset, C_1 = \emptyset$
Call Algorithm 7 to build a level-$L$ 2-HST $T$ based on input $U$
**for** *each node $v$ in $T$* **do**
   |   $N_v \leftarrow |U \cap T(v)|$
   |   $score(v) \leftarrow N_v \cdot 2^{h_v}$
**while** $|C_1| < k$ **do**
   |   Add top $(k - |C_1|)$ nodes with highest score in $T$ to $C_1$
   |   **for** *each $v \in C_1$* **do**
   |    |   $C_1 = C_1 \setminus \{v\}$, if $\exists\, v' \in C_1$ that $v'$ is a descendant of $v$
$C_0 = $ FIND-LEAF$(T, C_1)$
**Output:** Initial center set $C_0 \subseteq U$

---

**Algorithm 3:** FIND-LEAF $(T, C_1)$

---

**Input:** $T, C_1$
**Initialization:** $C_0 = \emptyset$
**for** *each node $v$ in $C_1$* **do**
   |   **while** *$v$ is not a leaf node* **do**
   |    |   $v \leftarrow \arg_w \max\{N_w, w \in ch(v)\}$, where $ch(v)$ denotes the children nodes of $v$
   |   Add $v$ to $C_0$
**Output:** Initial center set $C_0 \subseteq U$

---

root nodes of $k$ disjoint subtrees. For any node $w$ with $T(w) \cap C_1 = \emptyset$ and a node $v \in C_1$, if $w$ is not an ancestor of $v$, then $score(w) \leq score(v)$.

**Leaf search.** After we find $C_1$ the set of $k$ subtrees, the next step is to find the center in each subtree using Algorithm 3 ("FIND-LEAF"). We employ a greedy search strategy, by finding the child node with largest score level by level, until a leaf is found. This approach is intuitive since the diameter of the partition ball exponentially decays with the level. Therefore, we are in a sense focusing more and more on the region with higher density (i.e., with more data points).

The time complexity is given as below, considering the Euclidean space as an example.

**Proposition 3.1** (Complexity). *Algorithm 2 takes $O(dn \log n)$ time in the Euclidean space.*

**Remark 3.1.** *The complexity of HST initialization is in general comparable to the $O(dnk)$ of k-median++. Our algorithm would be faster if $k$, the number of centers, is larger. See Appendix B for a numerical comparison of the running time.*

### 3.3 Approximation Error of HST Initialization

Firstly, we show that the initial center set produced by Algorithm 2 is already a good approximation to the optimal $k$-median solution. Let $\rho^T(x, y) = d_T(x, y)$ denote the "2-HST metric" between $x$ and $y$ in the 2-HST $T$, where $d_T(x, y)$ is the tree distance between nodes $x$ and $y$ in $T$. In 2-HST, by Definition 3.1 and $\triangle = 2^L$, in the analysis we assume equivalently that the edge weight of the $i$-th level $2^{i-1}$. The crucial step of our analysis is to examine the approximation error in terms of the 2-HST metric, after which the error can be easily adapted to the general metrics by Lemma 3.2.

**Lemma 3.2.** *(Bartal, 1996). In a metric space $(U, \rho)$ with $|U| = n$ and diameter $\triangle$, it holds that $E[\rho^T(x, y)] = O(\min\{\log n, \log \triangle\})\rho(x, y)$. In $\mathbb{R}^d$, $E[\rho^T(x, y)] = O(d)\rho(x, y)$.*

Recall $C_0, C_1$ from Algorithm 2. We define

$$cost_k^T(U) = \sum_{y \in U} \min_{x \in C_0} \rho^T(x, y), \tag{3}$$

$$cost_k^{T'}(U, C_1) = \min_{|F \cap T(v)| = 1, \forall v \in C_1} \sum_{y \in U} \min_{x \in F} \rho^T(x, y), \tag{4}$$

$$OPT_k^T(U) = \min_{F \subset U, |F|=k} \sum_{y \in U} \min_{x \in F} \rho^T(x,y) \equiv \min_{C_1'} cost_k^{T'}(U, C_1'). \tag{5}$$

For simplicity, we will use $cost_k^{T'}(U)$ to denote $cost_k^{T'}(U, C_1)$. Here, $OPT_k^T$ (5) is the cost of the global optimal solution with 2-HST metric. The last equality in (5) holds because the optimal centers set can always located in $k$ disjoint subtrees, as each leaf only contain one point. (3) is the $k$-median cost with 2-HST metric of the output $C_0$ of Algorithm 2. (4) is the oracle cost after the subtrees are chosen. That is, it represents the optimal cost to pick one center from each subtree in $C_1$. Firstly, we bound the approximation error of the subtree search process.

**Lemma 3.3** (Subtree search). $cost_k^{T'}(U) \leq 5OPT_k^T(U)$.

Next, we show that the greedy leaf search (Algorithm 3) only has constant extra multiplicative error.

**Lemma 3.4** (Leaf search). $cost_k^T(U) \leq 2cost_k^{T'}(U)$.

Combining Lemma 3.3 and Lemma 3.4, we have the next Theorem.

**Theorem 3.5** (2-HST error). *Running Algorithm 2, we have $cost_k^T(U) \leq 10OPT_k^T(U)$.*

Thus, HST-initialization produces an $O(1)$ approximation to $OPT$ in the 2-HST metric. Define $cost_k(U)$ as (2) for our HST centers, and the optimal cost w.r.t. $\rho$ as

$$OPT_k(U) = \min_{|F|=k} \sum_{y \in U} \min_{x \in F} \rho(x,y). \tag{6}$$

We have the following result in the general metric space based on Lemma 3.2.

**Theorem 3.6.** *Running Algorithm 2 gives $E[cost_k(U)] = O(\min\{\log n, \log \triangle\})OPT_k(U)$.*

**Remark 3.2.** *In the Euclidean space, Makarychev et al. (2019) proved $O(\log k)$ random projections suffice for $k$-median to achieve $O(1)$ error. Thus, if $\triangle = O(d)$ (e.g., bounded data), by Lemma 3.2, there exists an algorithm with HST initialization that achieves $O(\log(\min\{d, k\}))$ error, which is better than $O(\log k)$ of $k$-median++ when $d$ is small.*

**NDP-HST Local Search.** We can apply standard local search starting from the HST initialization in Algorithm 1. We call it the NDP-HST ("NDP" stands for "Non-Differentially Private") method, with the following guarantee.

**Theorem 3.7.** *NDP-HST achieves $O(1)$ approximation in expected $O(k \log \min\{\log n, \log \triangle\})$ number of iterations for input in general metric space.*

Before ending this section, we remark that HST initialization and the analysis can be extended to $k$-means clustering analogously (see Appendix D). In a general metric space, $E[cost_{km}(U)] = O(\min\{\log n, \log \triangle\})^2 OPT_{km}(U)$ where $cost_{km}(U)$ is the optimal $k$-means cost.

## 4 HST INITIALIZATION WITH DIFFERENTIAL PRIVACY

In this section, we consider private HST initialization method. Recall in this setting, $U$ is the universe of data points, and $D \subset U$ is a demand set that needs to be clustered with privacy. Since $U$ is public, running initialization algorithms on $U$ would preserve the privacy of $D$. Yet, this might be too expensive, and in many cases one would probably want to incorporate some information about $D$ in the initialization, since $D$ could be a very imbalanced subset of $U$. For example, $D$ may only contain data points from one cluster, out of tens of clusters in $U$. In this case, initialization on $U$ is likely to pick initial centers in multiple clusters, which would not be helpful for clustering on $D$. Next, we show how our HST initialization can be easily combined with differential privacy that at the same time contains information about the demand set $D$, leading to improved approximation error (Theorem 4.3) and empirical clustering performance (Section 5).

Again, suppose $T$ is an $L = \log \triangle$-level 2-HST of universe $U$ in a general metric space. Denote $N_v = |T(v) \cap D|$ for a node point $v$. Our private HST initialization is similar to the non-private Algorithm 2. To gain privacy, we perturb $N_v$ by adding i.i.d. Laplace noise:

$$\hat{N}_v = N_v + Lap(2^{(L-h_v)}/\epsilon),$$

---

**Algorithm 4:** HST initialization with differential privacy

---

**Input:** $U, D, \triangle, k, \epsilon$
Build a level-$L$ 2-HST $T$ based on input $U$
**for** *each node $v$ in $T$* **do**
$\quad$ $N_v \leftarrow |D \cap T(v)|$
$\quad$ $\hat{N}_v \leftarrow N_v + Lap(2^{(L-h_v)}/\epsilon)$
$\quad$ $score(v) \leftarrow \hat{N}(v) \cdot 2^{h_v}$
Based on $\hat{N}_v$, apply the same strategy as Algorithm 2: find $C_1$; $C_0 = $ FIND-LEAF($C_1$)
**Output:** Private initial center set $C_0 \subseteq U$

---

where $Lap(2^{(L-h_v)}/\epsilon)$ is a Laplace random number with rate $2^{(L-h_v)}/\epsilon$. We will use the perturbed $\hat{N}_v$ for node sampling instead of the true value $N_v$, as described in Algorithm 4. The DP guarantee of this initialization scheme is straightforward by the composition of the Laplace mechanisms.

**Theorem 4.1.** *Algorithm 4 is $\epsilon$-differentially private.*

*Proof.* For each level $i$, the subtrees $T(v, i)$ are disjoint to each other. The privacy used in $i$-th level is $\epsilon/2^{(L-i)}$, and the total privacy is $\sum_i \epsilon/2^{(L-i)} < \epsilon$. $\qquad\square$

We now consider the approximation error. As the structure of the analysis is similar to the non-DP case, we present the main result here and defer the detailed proofs to Appendix C.

**Theorem 4.2.** *Algorithm 4 outputs an initial center set such that*

$$E[cost_k(D)] = O(\log n)(OPT_k(D) + k\epsilon^{-1}\triangle \log n).$$

**DP-HST Local Search.** Similarly, we can use private HST initialization to improve the performance of private $k$-median local search, which is presented in Algorithm 5. After initialization, the DP local search procedure follows Gupta et al. (2010) using the exponential mechanism.

---

**Algorithm 5:** DP-HST local search

---

**Input:** $U$, demand points $D \subseteq U$, parameter $k, \epsilon, T$
**Initialization:** $F_1$ the private initial centers generated by Algorithm 4 with privacy $\epsilon/2$
Set parameter $\epsilon' = \frac{\epsilon}{2\triangle(T+1)}$
**for** $i = 1$ *to* $T$ **do**
$\quad$ Select $(x, y) \in F_i \times (V \setminus F_i)$ with prob. proportional to $\exp(-\epsilon' \times (cost(F_i - \{x\} + \{y\})))$
$\quad$ Let $F_{i+1} \leftarrow F_i - \{x\} + \{y\}$
Select $j$ from $\{1, 2, ..., T+1\}$ with probability proportional to $\exp(-\epsilon' \times cost(F_j))$
**Output:** $F = F_j$ the private center set

---

**Theorem 4.3.** *Algorithm 5 achieves $\epsilon$-differential privacy. The output centers admit*

$$cost_k(D) \leq 6OPT_k(D) + O(\epsilon^{-1}k^2\triangle(\log\log n)\log n)$$

*with probability $(1 - 1/poly(n))$, with $T = O(k\log\log n)$ iterations.*

The DP local search with random initialization (Gupta et al., 2010) has 6 multiplicative error and $O(\epsilon^{-1}\triangle k^2 \log^2 n)$ additive error. Our result improves the $\log n$ term to $\log\log n$ in the additive error. Meanwhile, the number of iterations needed is improved from $T = O(k\log n)$ to $O(k\log\log n)$ (see Appendix B for an empirical justification). Notably, it has been shown in Gupta et al. (2010) that for $k$-median problem, the lower bounds on the multiplicative and additive error of any $\epsilon$-DP algorithm are $O(1)$ and $O(\epsilon^{-1}\triangle k \log(n/k))$, respectively. Our result matches the lower bound on the multiplicative error, and the additive error is only worse than the bound by a factor of $O(k\log\log n)$ which is typically small in many cases. Thus, our private HST initialization method pushes the approximation error of private local search closer to the lower bound. To our knowledge, Theorem 4.3 is the first result in literature to improve the error of DP local search in general metric space.

## 5 EXPERIMENTS

We numerically test the proposed methods on two problems—clustering in an Euclidean space and on a graph. Our results show that the proposed HST initialization can improve the performance of using $k$-median++ initialization in both non-private and private clustering tasks.

### 5.1 DATASETS AND ALGORITHMS

**Discrete Euclidean space.** Following Balcan et al. (2017), we test $k$-median clustering on the MNIST hand-written digit dataset (LeCun et al., 1998) with 10 natural clusters (digit 0 to 9). We set $U$ as 10000 randomly chosen data points. We choose the demand set $D$ using two strategies: 1) "balance", where we randomly choose 500 samples from $U$; 2) "imbalance", where $D$ contains 500 random samples from $U$ only from digit "0" and "8" (two clusters). We note that, the imbalanced $D$ is a very practical setting in real-world scenarios, where data are typically not uniformly distributed. On this dataset, we test clustering with both $l_1$ and $l_2$ distance as the underlying metric.

**Metric space induced by graph.** Random graphs have been widely considered in testing $k$-median methods (Balcan et al., 2013; Todo et al., 2019). The construction of graphs follows a similar approach as the synthetic *pmedinfo* graphs provided by the popular OR-Library (Beasley, 1990). The metric $\rho$ for this experiment is the shortest (weighted) path distance on graph. To generate a size $n$ graph, we first randomly split the nodes into 10 clusters. Within each cluster, each pair of nodes is connected with probability 0.2 and weight drawn from standard uniform distribution. For each pair of clusters, we randomly connect some nodes from each cluster, with weights following uniform $[0.5, r]$. A larger $r$ makes the graph more separable, i.e., clusters are farther from each other (see Appendix B for example graphs). We present two cases: $r = 1$ and $r = 100$. For this task, $U$ has 3000 nodes, and the private set $D$ is chosen using similar "balanced" and "imbalanced" scheme as described above. In the imbalanced case, we choose $D$ randomly from only two clusters.

**Algorithms.** We compare the following clustering algorithms in both non-DP and DP setting.

- **NDP-rand:** Local search on $D$ with random initialization (Algorithm 1).
- **NDP-kmedian++:** Algorithm 1 with $k$-median++ initialization.
- **NDP-HST:** Algorithm 1 with HST initialization, as described in Section 3.
- **DP-rand:** Standard private local search algorithm (Gupta et al., 2010), which is Algorithm 5 with initial centers randomly chosen from $U$.
- **DP-kmedian++:** Algorithm 5 with $k$-median++ initialization run on $U$.
- **DP-HST:** Private local search with HST-initialization (Algorithm 5). For non-private tasks, we set $L = 6$. For private clustering, we use $L = 8$.

For non-DP methods, we set $\alpha = 10^{-3}$ in Algorithm 1 and the maximum number of iterations as 20. To examine the quality of initialization as well as the final centers, We report both the cost at initialization and the cost of the final output. For DP methods, we run the algorithms for $T = 20$ steps and report the results with $\epsilon = 1$. We test $k \in \{2, 5, 10, 15, 20\}$. The average cost over $T$ iterations is reported for more robustness. All results are averaged over 10 independent repetitions.

### 5.2 RESULTS

The results on MNIST dataset are given in Figure 2 for $l_1$ (left two columns) and $l_2$ (right two columns) metric. The comparisons are similar in both cases.

- We see that the initial centers generated by HST has lower $k$-median cost than $k$-median++ and random initialization, for both non-DP and DP setting, and for both balanced and imbalanced demand set $D$. This confirms that the proposed HST initialization is more powerful than $k$-median++ in finding good initial centers.
- From the final $k$-median cost plots, we also observe lower cost of HST approaches in DP clustering. In the non-DP case, the curves overlap, which means despite that HST offers better initial centers, local search can always find a good solution eventually.

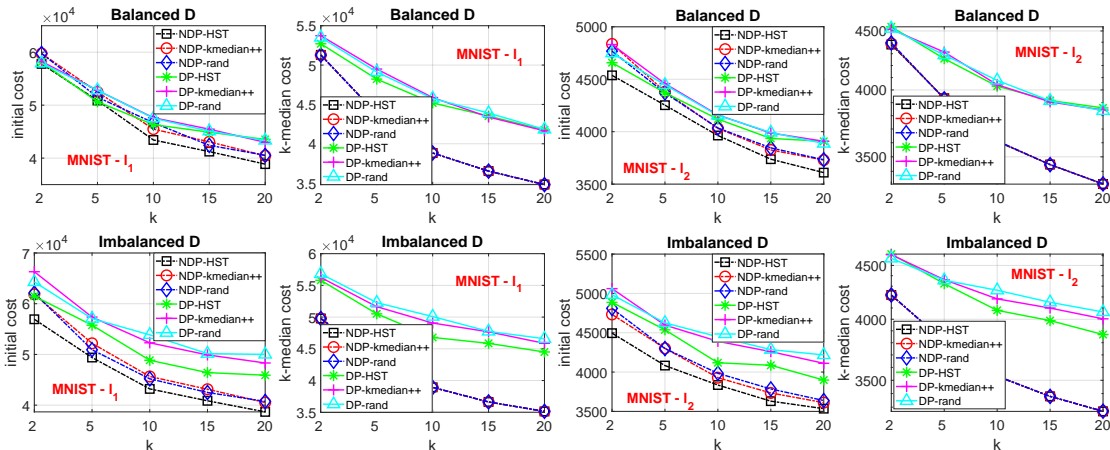

Figure 2: $k$-median cost on MNIST dataset. **1st row:** balanced $D$. **2nd row:** imbalanced $D$. **Column 1 & 3:** initial cost. **Column 2 & 4:** final $k$-median cost.

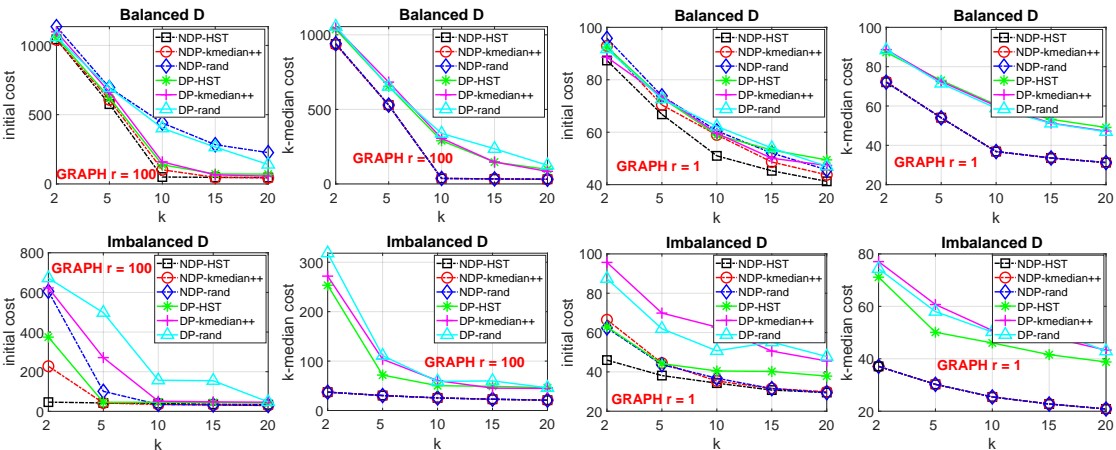

Figure 3: $k$-median costs on graph dataset. **1st row:** balanced $D$. **2nd row:** imbalanced $D$.

- The advantage of HST in both initial and final cost is more significant when $D$ is an imbalanced subset of $U$. As mentioned before, this is because our HST initialization approach also privately incorporates the information of $D$.

The set of results on the graph dataset is reported in Figure 3, which gives us similar conclusion. In all cases, our proposed HST method finds better initial centers with smaller cost than $k$-median++. In terms of the final clustering output, HST again considerably outperforms $k$-median++ in the private and imbalanced $D$ setting, for both $r = 100$ (highly separable graph) and $r = 1$ (less separable graph). The improvement of HST over $k$-median++ is especially significant in the harder tasks when $r = 1$, i.e., the clusters are nearly mixed up.

## 6    CONCLUSION

In this paper, we propose a new initialization framework for the $k$-median problem in general discrete metric space. Our approach is called HST, which leverages tools from metric embedding theory. Our novel tree search approach has comparable efficiency and cost to $k$-median++ initialization. Moreover, we propose differentially private (DP) HST center initialization algorithm, which adapts to the private demand point set, leading to better clustering performance. When combined with subsequent DP local search heuristic, our algorithm is able to improve the addition error of prior works, which is close to the theoretical lower bound within a small factor. Experiments with Euclidean metrics and graph metrics verify the effectiveness of our methods.

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

# A    POSTPONED ALGORITHMS

## A.1    $k$-MEDIAN++

In the paper, we compared our HST initialization mainly with another (perhaps most well-known) initialization algorithm for clustering, the $k$-median++ (Arthur & Vassilvitskii, 2007). For reference, we present the concrete procedures in Algorithm 6. Here, the function $D(u, C)$ is the shortest distance from a data point $u$ to the closest (center) point in set $C$. Arthur & Vassilvitskii (2007) showed that the output centers $C$ by $k$-median++ achieves $O(\log k)$ approximation error, in $O(dnk)$ time.

---

**Algorithm 6:** $k$-median++ (Arthur & Vassilvitskii, 2007)

**Input:** Data points $U$, number of centers $k$
Set $C = [\,]$
Randomly pick a point $c_1$ in $U$ and set $C = C \cup \{c_1\}$
**for** $i = 2$ *to* $k$ **do**
    Select $c_i = u \in U$ with probability $\frac{D(u,C)}{\sum_{u' \in U} D(u',C)}$
    $C = C \cup \{c_i\}$
**Output:** $k$-median++ initial center set $C$

---

## A.2    CONSTRUCTING A 2-HST

As presented in Algorithm 7, the construction starts by applying a permutation $\pi$ on $U$, such that in following steps the points are picked in a random sequence. We first find a padded decomposition $P_L = \{P_{L,1}, ..., P_{L,n_L}\}$ of $U$ with parameter $\beta = \triangle/2$. The center of each partition in $P_{L,j}$ serves as a root node in level $L$. Then, we re-do a padded decomposition for each partition $P_{L,j}$, to find sub-partitions with diameter $\beta = \triangle/4$, and set the corresponding centers as the nodes in level $L-1$, and so on. Each partition at level $i$ is obtained with $\beta = \triangle/2^{L-i}$. This process proceeds until a node has a single point, or a pre-specified tree depth is reached. In Figure 1, we provide an example of $L = 3$-level 2-HST (left panel), along with its underlying padded decompositions (right panel).

---

**Algorithm 7:** Build 2-HST$(U, L)$

**Input:** Data points $U$ with diameter $\triangle$, $L$
Randomly pick a point in $U$ as the root node of $T$
Let $r = \triangle/2$
Apply a permutation $\pi$ on $U$ `// so points will be chosen in a random sequence`
**for** *each* $v \in U$ **do**
    Set $C_v = [v]$
    **for** *each* $u \in U$ **do**
        Add $u \in U$ to $C_v$ if $d(v, u) \le r$ and $u \notin \bigcup_{v' \neq v} C_{v'}$
Set the non-empty clusters $C_v$ as the children nodes of $T$
**for** *each non-empty cluster* $C_v$ **do**
    Run 2-HST$(C_v, L - 1)$ to extend the tree $T$; stop until $L$ levels or reaching a leaf node
**Output:** 2-HST $T$

---

# B MORE EXPERIMENTS

## B.1 EXAMPLES OF GRAPH DATA

In Figure 4, we plot two example graphs (subgraphs of 50 nodes) with $r = 100$ and $r = 1$. When $r = 100$, the graph is highly separable (i.e., clusters are far from each other). When $r = 1$, the clusters are harder to be distinguished from each other.

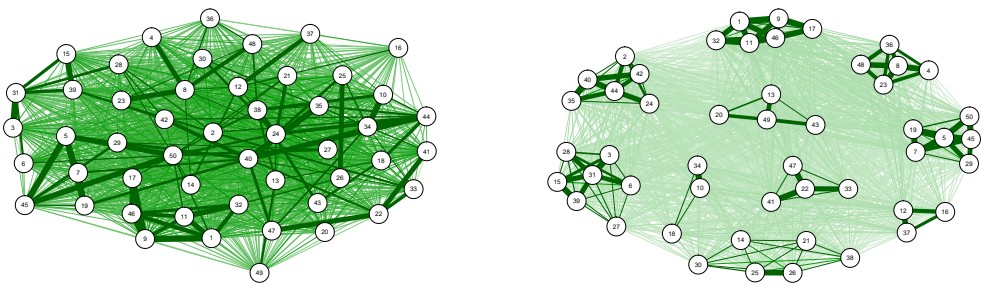

Figure 4: Example of synthetic graphs: subgraph of 50 nodes. **Left:** $r = 1$. **Right:** $r = 100$. Darker and thicker edged have smaller distance. When $r = 100$, the graph is more separable.

## B.2 RUNNING TIME COMPARISON WITH $k$-MEDIAN++

In Proposition 3.1, we show that our HST initialization algorithm admits $O(dn \log n)$ complexity when considering the Euclidean space. With a smart implementation of Algorithm 6 where each data point tracks its distance to the current closest candidate center in $C$, $k$-median++ has $O(dnk)$ running time. Therefore, the running time of our algorithm is in general comparable to $k$-median++. Our method would run faster if $k = \Omega(\log n)$. In Figure 5, we plot the empirical running time of HST initialization against $k$-median++, on MNIST dataset with $l_2$ distance (similar comparison holds for $l_1$). From the left subfigure, we see that $k$-median++ becomes slower with increasing $k$, and our method is more efficient when $k > 20$. In the right panel, we observe that the running time of both methods increases with larger sample size $n$. Our HST algorithm has a slightly faster increasing rate, which is predicted by the complexity comparison ($n \log n$ v.s. $n$). However, this difference in $\log n$ factor would not be too significant unless the sample size is extremely large. Overall, our numerical results suggest that in general, the proposed HST initialization would have similar efficiency as $k$-median++ in common practical scenarios.

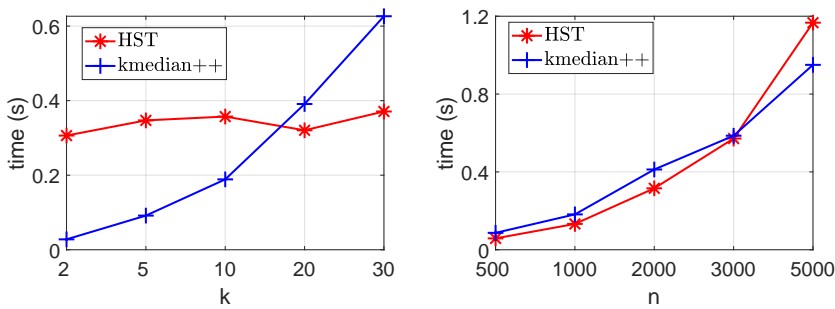

Figure 5: Empirical time comparision of HST initialization v.s. $k$-median++, on MNIST dataset with $l_2$ distance. **Left:** The running time against $k$, on a subset of $n = 2000$ data points. **Right:** The running time against $n$, with $k = 20$ centers.

### B.3 IMPROVED ITERATION COST OF DP-HST

In Theorem 4.3, we show that under differential privacy constraints, the proposed DP-HST (Algorithm 5) improves both the approximation error and the number of iterations required to find a good solution of classical DP local search (Gupta et al., 2010). In this section, we provide some numerical results to justify the theory.

First, we need to properly measure the iteration cost of DP local search. This is because, unlike the non-private clustering, the $k$-median cost after each iteration in DP local search is not decreasing monotonically, due to the probabilistic exponential mechanism. To this end, for the cost sequence with length $T = 20$, we compute its moving average sequence with window size 5. Attaining the minimal value of the moving average indicates that the algorithm has found a "local optimum", i.e., it has reached a "neighborhood" of solutions with small clustering cost. Thus, we use the number of iterations to reach such local optimum as the measure of iteration cost. The results are provided in Figure 6. We see that on all the tasks (MNIST with $l_1$ and $l_2$ distance, and graph dataset with $r = 1$ and $r = 100$), DP-HST has significantly smaller iterations cost. In Figure 7, we further report the $k$-median cost of the best solution in $T$ iterations found by each DP algorithm. We see that DP-HST again provide the smallest cost. This additional set of experiments again validates the claims of Theorem 4.3, that DP-HST is able to found better initial centers in fewer iterations.

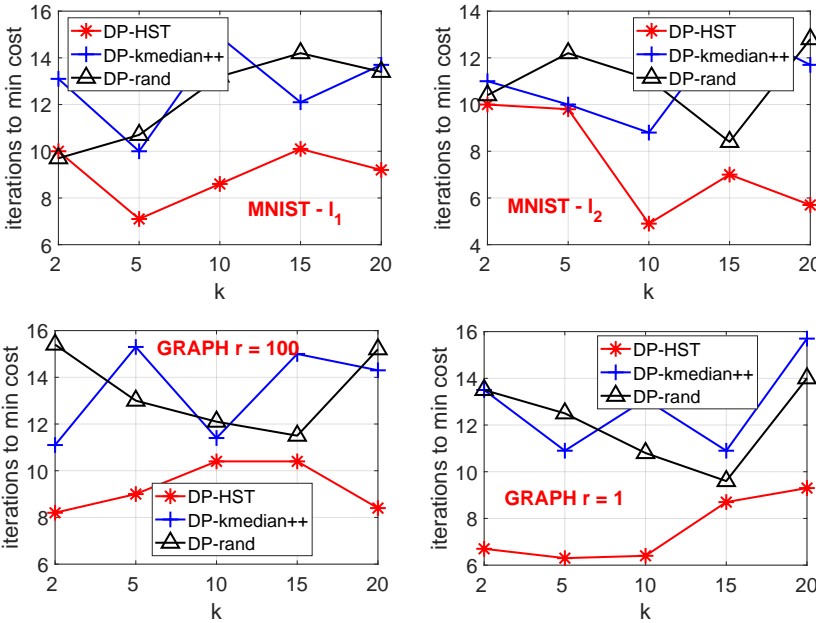

Figure 6: Iteration cost to reach a locally optimal solution, on MNIST and graph datasets with different $k$. The demand set is an imbalanced subset of the universe.

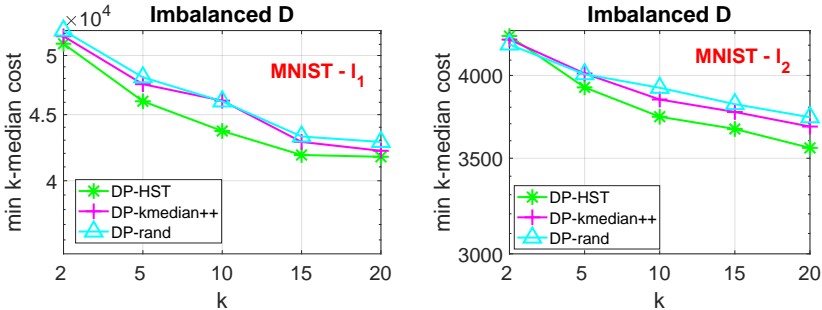

Figure 7: The $k$-median cost of the best solution found by each differentially private algorithm. The demand set is an imbalanced subset of the universe. Same comparison holds on graph data.

# C PROOFS

The following composition result of differential privacy will be used in our proof.

**Theorem C.1** (Composition Theorem (Dwork, 2006)). *If Algorithms $\mathbb{A}_1, \mathbb{A}_2, ..., \mathbb{A}_m$ are $\epsilon_1, \epsilon_2, ..., \epsilon_m$ differentially private respectively, then the union $(\mathbb{A}_1(D), \mathbb{A}_2(D), ..., \mathbb{A}_m(D))$ is $\sum_{i=1}^{m} \epsilon_i$-DP.*

## C.1 PROOF OF LEMMA 3.3

*Proof.* Consider the intermediate output of Algorithm 2, $C_1 = \{v_1, v_2, ..., v_k\}$, which is the set of roots of the minimal subtrees each containing exactly one output center $C_0$. Suppose one of the optimal "root set" that minimizes (4) is $C_1^* = \{v_1', v_2', ..., v_k'\}$. If $C_1 = C_1^*$, the proof is done. Thus, we prove the case for $C_1 \neq C_1^*$. Note that $T(v), v \in C_1$ are disjoint subtrees. We have the following reasoning.

- Case 1: for some $i, j'$, $v_i$ is a descendant node of $v_j'$. Since the optimal center point $f^*$ is a leaf node by the definition of (4), we know that there must exist one child node of $v_j'$ that expands a subtree which contains $f^*$. Therefore, we can always replace $v_j'$ by one of its child nodes. Hence, we can assume that $v_i$ is not a descendant of $v_j'$.

  Note that, we have $score(v_j') \leq score(v_i)$ if $v_j' \notin C_1^* \cap C_1$. Algorithm 2 sorts all the nodes based on cost value, and it would have more priority to pick $v_j'$ than $v_i$ if $score(v_j') > score(v_i)$ and $v_i$ is not a child node of $v_j'$.

- Case 2: for some $i, j'$, $v_j'$ is a descendant of $v_i$. In this case, optimal center point $f^*$, which is a leaf of $T(v_i)$, must also be a leaf node of $T(v_j')$. We can simply replace $C_1$ with the swap $C_1 \setminus \{v_i\} + \{v_j'\}$ which does not change $cost_k^{T'}(U)$. Hence, we can assume that $v_j'$ is not a descendant of $v_i$.

- Case 3: Otherwise. By the construction of $C_1$, we know that $score(v_j') \leq \min\{score(v_i), i = 1, ..., k\}$ when $v_j' \in C_1^* \setminus C_1$. Consider the swap between $C_1$ and $C_1^*$. By the definition of tree distance, we have $OPT_k^T(U) \geq \sum_{v_i \in C_1 \setminus C_1^*} N_{v_i} 2^{h_{v_i}}$, since $\{T(v_i), v_i \in C_1 \setminus C_1^*\}$ does not contain any center of the optimal solution determined by $C_1^*$ (which is also the optimal "root set" for $OPT_k^T(U)$).

Thus, we only need to consider Case 3. Let us consider the optimal clustering with center set be $C^* = \{c_1^*, c_2^*, ..., c_k^*\}$ (each center $c_j^*$ is a leaf of subtree whose root be $c_j'$), and $S_j'$ be the leaves assigned to $c_j^*$. Let $S_j$ denote the set of leaves in $S_j'$ whose distance to $c_j^*$ is strictly smaller than its distance to any centers in $C_1$. Let $P_j$ denote the union of paths between leaves of $S_j$ to its closest center in $C_1$. Let $v_j''$ be the nodes in $P_j$ with highest level satisfying $T(v_j'') \cap C_1 = \emptyset$. The score of $v_j''$ is $2^{h_{v_j''}} N(v_j'')$. That means the swap with a center $v_j'$ into $C_1$ can only reduce $4 \cdot 2^{h_{v_j''}} N(v_j'')$ to $cost_k^{T'}(U)$ (the tree distance between any leaf in $S_j$ and its closest center in $C_1$ is at most $4 \cdot 2^{h_{v_j''}}$). We just use $v_j'$ to represent $v_j''$ for later part of this proof for simplicity. By our reasoning, summing all the swaps over $C_1^* \setminus C_1$ gives

$$cost_k^{T'}(U) - OPT_k^T(U) \leq 4 \sum_{v_j' \in C_1^* \setminus C_1} N_{v_j'} 2^{h_{v_j'}},$$

$$OPT_k^T(U) \geq \sum_{v_i \in C_1 \setminus C_1^*} N_{v_i} 2^{h_{v_i}}.$$

Also, based on our discussion on Case 1, it holds that

$$N_{v_j'} 2^{h_{v_j'}} - N_{v_i} 2^{h_{v_i}} \leq 0.$$

Summing them together, we have $cost_k^{T'}(U) \leq 5 OPT_k^T(U)$. □

## C.2  PROOF OF LEMMA 3.4

*Proof.* Since the subtrees in $C_1$ are disjoint, it suffices to consider one subtree with root $v$. With a little abuse of notation, let $cost_1^{T'}(v, U)$ denote the optimal $k$-median cost within the point set $T(v)$ with one center in 2-HST:

$$cost_1^{T'}(v, U) = \min_{x \in T(v)} \sum_{y \in T(v)} \rho^T(x, y), \tag{7}$$

which is the optimal cost within the subtree. Suppose $v$ has more than one children $u, w, ...$, otherwise the optimal center is clear. Suppose the optimal solution of $cost_1^{T'}(v, U)$ chooses a leaf node in $T(u)$, and our HST initialization algorithm picks a leaf of $T(w)$. If $u = w$, then HST chooses the optimal one where the argument holds trivially. Thus, we consider $u \neq w$. We have the following two observations:

- Since one needs to pick a leaf of $T(u)$ to minimize $cost_1^{T'}(v, U)$, we have $cost_1^{T'}(v, U) \geq \sum_{x \in ch(v), x \neq u} N_x \cdot 2^{h_x}$ where $ch(u)$ denotes the children nodes of $u$.

- By our greedy strategy, $cost_1^T(v, U) \leq \sum_{x \in ch(u)} N_x \cdot 2^{h_x} \leq cost_1^{T'}(v, U) + N_u \cdot 2^{h_u}$.

Since $h_u = h_w$, we have

$$2^{h_u} \cdot (N_u - N_w) \leq 0,$$

since our algorithm picks subtree roots with highest scores. Then we have $cost_1^T(v, U) \leq cost_1^{T'}(v, U) + N_w \cdot 2^{h_w} \leq 2cost_1^{T'}(v, U)$. Since the subtrees in $C_1$ are disjoint, the union of centers for $OPT_1^T(v, U)$, $v \in C_1$ forms the optimal centers with size $k$. Note that, for any data point $p \in U \setminus C_1$, the tree distance $\rho^T(p, f)$ for $\forall f$ that is a leaf node of $T(v)$, $v \in C_1$ is the same. That is, the choice of leaf in $T(v)$ as the center does not affect the $k$-median cost under 2-HST metric. Therefore, union bound over $k$ subtree costs completes the proof. $\square$

## C.3  PROOF OF PROPOSITION 3.1

*Proof.* It is known that the 2-HST can be constructed in $O(dn \log n)$ (Bartal, 1996). The subtree search in Algorithm 2 involves at most sorting all the nodes in the HST based on the score, which takes $O(n log n)$. We use a priority queue to store the nodes in $C_1$. When we insert a new node $v$ into queue, its parent node (if existing in the queue) would be removed from the queue. The number of nodes is $O(n)$ and each operation (insertion, deletion) in a priority queue based on score has $O(\log n)$ complexity. Lastly, the total time to obtain $C_0$ is $O(n)$, as the FIND-LEAF only requires a top down scan in $k$ disjoint subtrees of $T$. Summing parts together proves the claim. $\square$

## C.4  PROOF OF THEOREM 4.2

Similarly, we prove the error in general metric by first analyzing the error in 2-HST metric. Then the result follows from Lemma 3.2. Let $cost_k^T(D)$, $cost_k^{T'}(D)$ and $OPT_k^T(D)$ be defined analogously to (3), (4) and (5), where "$y \in U$" in the summation is changed into "$y \in D$" since $D$ is the demand set. That is,

$$cost_k^T(D) = \sum_{y \in D} \min_{x \in C_0} \rho^T(x, y), \tag{8}$$

$$cost_k^{T'}(D, C_1) = \min_{|F \cap T(v)| = 1, \forall v \in C_1} \sum_{y \in D} \min_{x \in F} \rho^T(x, y), \tag{9}$$

$$OPT_k^T(D) = \min_{F \subset D, |F| = k} \sum_{y \in D} \min_{x \in F} \rho^T(x, y) \equiv \min_{C_1'} cost_k^{T'}(D, C_1'). \tag{10}$$

We have the following.

**Lemma C.2.** $cost_k^T(D) \leq 10 OPT_k^T(D) + 10ck\epsilon^{-1}\triangle \log n$ *with probability* $1 - 4k/n^c$.

*Proof.* The result follows by combining the following Lemma C.4, Lemma C.5, and applying union bound. □

**Lemma C.3.** *For any node $v$ in $T$, with probability $1 - 1/n^c$, $|\hat{N}_v \cdot 2^{h_v} - N_v \cdot 2^{h_v}| \leq c\epsilon^{-1}\triangle \log n$.*

*Proof.* Since $\hat{N}_v = N_v + Lap(2^{(L-h_v)/2}/\epsilon)$, we have

$$Pr[|\hat{N}_v - N_v| \geq x/\epsilon] = exp(-x/2^{(L-h_v)}).$$

As $L = \log \triangle$, we have

$$Pr[|\hat{N}_v - N_v| \geq x\triangle/(2^{h_v}\epsilon)] \leq exp(-x).$$

Hence, for some constant $c > 0$,

$$Pr[|\hat{N}_v \cdot 2^{h_v} - N_v \cdot 2^{h_v}| \leq c\epsilon^{-1}\triangle \log n] \geq 1 - exp(-c\log n) = 1 - 1/n^c.$$

□

**Lemma C.4** (DP Subtree Search)**.** *With probability $1 - 2k/n^c$, $cost_k^{T'}(D) \leq 5OPT_k^T(D) + 4ck\epsilon^{-1}\triangle \log n$.*

*Proof.* The proof is similar to that of Lemma 3.3. Consider the intermediate output of Algorithm 2, $C_1 = \{v_1, v_2, ..., v_k\}$, which is the set of roots of the minimal disjoint subtrees each containing exactly one output center $C_0$. Suppose one of the optimal "root set" that minimizes (4) is $C_1^* = \{v_1', v_2', ..., v_k'\}$. Assume $C_1 \neq C_1^*$. By the same argument as the proof of Lemma 3.3, we consider for some $i, j$ such that $v_i \neq v_j'$, where $v_i$ is not a descendent of $v_j'$ and $v_j'$ is either a descendent of $v_i$. By the construction of $C_1$, we know that $score(v_j') \leq \min\{score(v_i), i = 1, ..., k\}$ when $v_j' \in C_1^* \setminus C_1$. Consider the swap between $C_1$ and $C_1^*$. By the definition of tree distance, we have $OPT_k^T(U) \geq \sum_{v_i \in C_1 \setminus C_1^*} N_{v_i} 2^{h_{v_i}}$, since $\{T(v_i), v_i \in C_1 \setminus C_1^*\}$ does not contain any center of the optimal solution determined by $C_1^*$ (which is also the optimal "root set" for $OPT_k^T$). Let us consider the optimal clustering with center set be $C^* = \{c_1^*, c_2^*, ..., c_k^*\}$ (each center $c_j^*$ is a leaf of subtree whose root be $c_j'$), and $S_j'$ be the leaves assigned to $c_j^*$. Let $S_j$ denote the set of leaves in $S_j'$ whose distance to $c_j^*$ is strictly smaller than its distance to any centers in $C_1$. Let $P_j$ denote the union of paths between leaves of $S_j$ to its closest center in $C_1$. Let $v_j''$ be the nodes in $P_j$ with highest level satisfying $T(v_j'') \cap C_1 = \emptyset$. The score of $v_j''$ is $2^{h_{v_j''}}N(v_j'')$. That means the swap with a center $v_j'$ into $C_1$ can only reduce $4 \cdot 2^{h_{v_j''}}N(v_j'')$ to $cost_k^{T'}(U)$ (the tree distance between any leaf in $S_j$ and its closest center in $C_1$ is at most $4 \cdot 2^{h_{v_j''}}$). We just use $v_j'$ to represent $v_j''$ for later part of this proof for simplicity. Summing all the swaps over $C_1^* \setminus C_1$, we obtain

$$cost_k^{T'}(U) - OPT_k^T(U) \leq 4\sum_{v_j' \in C_1^* \setminus C_1} N_{v_j'} 2^{h_{v_j'}},$$

$$OPT_k^T(U) \geq \sum_{v_i \in C_1 \setminus C_1^*} N_{v_i} 2^{h_{v_i}}.$$

Applying union bound with Lemma C.3, with probability $1 - 2/n^c$, we have

$$N_{v_j'} 2^{h_{v_j'}} - N_{v_i} 2^{h_{v_i}} \leq 2c\epsilon^{-1}\triangle \log n.$$

Consequently, we have with probability, $1 - 2k/n^c$,

$$cost_k^{T'}(D) \leq 5OPT_k^T(D) + 4c|C_1 \setminus C_1^*|\epsilon^{-1}\triangle \log n$$
$$\leq 5OPT_k^T(D) + 4ck\epsilon^{-1}\triangle \log n.$$

□

**Lemma C.5** (DP Leaf Search)**.** *With probability $1 - 2k/n^c$, Algorithm 4 produces initial centers with $cost_k^T(D) \leq 2cost_k^{T'}(D) + 2ck\epsilon^{-1}\triangle \log n$.*

*Proof.* The proof strategy follows Lemma 3.4. We first consider one subtree with root $v$. Let $cost_1^{T'}(v, U)$ denote the optimal $k$-median cost within the point set $T(v)$ with one center in 2-HST:

$$cost_1^{T'}(v, D) = \min_{x \in T(v)} \sum_{y \in T(v) \cap D} \rho^T(x, y). \tag{11}$$

Suppose $v$ has more than one children $u, w, ...,$ and the optimal solution of $cost_1^{T'}(v, U)$ chooses a leaf node in $T(u)$, and our HST initialization algorithm picks a leaf of $T(w)$. If $u = w$, then HST chooses the optimal one where the argument holds trivially. Thus, we consider $u \neq w$. We have the following two observations:

- Since one needs to pick a leaf of $T(u)$ to minimize $cost_1^{T'}(v, U)$, we have $cost_1^{T'}(v, U) \geq \sum_{x \in ch(v), x \neq u} N_x \cdot 2^{h_x}$ where $ch(u)$ denotes the children nodes of $u$.

- By our greedy strategy, $cost_1^T(v, U) \leq \sum_{x \in ch(u)} N_x \cdot 2^{h_x} \leq cost_1^{T'}(v, U) + N_u \cdot 2^{h_u}$.

As $h_u = h_w$, leveraging Lemma C.3, with probability $1 - 2/n^c$,

$$2^{h_u} \cdot (N_u - N_w) \leq 2^{h_u}(\hat{N}_u - \hat{N}_w) + 2c\epsilon^{-1} \triangle \log n$$
$$\leq 2c\epsilon^{-1} \triangle \log n.$$

since our algorithm picks subtree roots with highest scores. Then we have $cost_1^T(v, D) \leq cost_k^{T'}(v, D) + N_w \cdot 2^{h_u} + 2c\epsilon^{-1} \triangle \log n \leq 2cost_k^{T'}(v, D) + 2c\epsilon^{-1} \triangle \log n$ with high probability. Lastly, applying union bound over the disjoint $k$ subtrees gives the desired result. □

## C.5 PROOF OF THEOREM 4.3

*Proof.* The privacy analysis is straightforward, by using the composition theorem (Theorem C.1). Since the sensitivity of $cost(\cdot)$ is $\triangle$, in each swap iteration the privacy budget is $\epsilon/2(T+1)$. Also, we spend another $\epsilon/2(T+1)$ privacy for picking a output. Hence, the total privacy is $\epsilon/2$ for local search. Algorithm 4 takes $\epsilon/2$ DP budget for initialization, so the total privacy is $\epsilon$.

The analysis of the approximation error follows from Gupta et al. (2010), where the initial cost is reduced by our private HST method. We need the following two lemmas.

**Lemma C.6** (Gupta et al. (2010)). *Assume the solution to the optimal utility is unique. For any output $o \in O$ of $2\triangle\epsilon$-DP exponential mechanism on dataset $D$, it holds for $\forall t > 0$ that*

$$Pr[q(D, o) \leq \max_{o \in O} q(D, o) - (\ln |O| + t)/\epsilon] \leq e^{-t},$$

*where $|O|$ is the size of the output set.*

**Lemma C.7** (Arya et al. (2004)). *For any set $F \subseteq D$ with $|F| = k$, there exists some swap $(x, y)$ such that the local search method admits*

$$cost_k(F, D) - cost_k(F - \{x\} + \{y\}, D) \geq \frac{cost_k(F, D) - 5OPT(D)}{k}.$$

From Lemma C.7, we know that when $cost_k(F_i, D) > 6OPT(D)$, there exists a swap $(x, y)$ s.t.

$$cost_k(F_i - \{x\} + \{y\}, D) \leq (1 - \frac{1}{6k})cost_k(F_i, D).$$

At each iteration, there are at most $n^2$ possible outputs (i.e., possible swaps), i.e., $|O| = n^2$. Using Lemma C.6 with $t = 2 \log n$, for $\forall i$,

$$Pr[cost_k(F_{i+1}, D) \geq cost_k(F_{i+1}^*, D) + 4\frac{\log n}{\epsilon'}] \geq 1 - 1/n^2,$$

where $cost_k(F_{i+1}^*, D)$ is the minimum cost among iteration $1, 2, ..., t+1$. Hence, we have that as long as $cost(F_i, D) > 6OPT(D) + \frac{24k \log n}{\epsilon'}$, the improvement in cost is at least by a factor of

$(1 - \frac{1}{6k})$. By Theorem 4.2, we have $cost_k(F_1, D) \leq C(\log n)(6OPT(D) + 6k\triangle \log n/\epsilon)$ for some constant $C > 0$. Let $T = 6Ck \log \log n$. We have that

$$E[cost(F_i, D)] \leq (6OPT(D) + 6k\epsilon^{-1}\triangle \log n)C(\log n)(1 - 1/6k)^{6Ck \log \log n}$$

$$\leq 6OPT(D) + 6k\epsilon^{-1}\triangle \log n \leq 6OPT(D) + \frac{24k \log n}{\epsilon'}.$$

Therefore, with probability at least $(1 - T/n^2)$, there exists an $i <= T$ s.t. $cost(F_i, D) \leq 6OPT(D) + \frac{24k \log n}{\epsilon'}$. Then by using the Lemma C.7, one will pick an $F_j$ with additional additive error $4 \ln n/\epsilon'$ to the $\min\{cost(F_j, D), j = 1, 2, ..., T\}$ with probability $1 - 1/n^2$. Consequently, we know that the expected additive error is

$$24k\triangle \log n/\epsilon' + 4\log n/\epsilon' = O(\epsilon^{-1}k^2\triangle(\log \log n)\log n),$$

with probability $1 - 1/poly(n)$.

$\square$

# D   EXTEND HST INITIALIZATION TO $k$-MEANS

Naturally, our HST method can also be applied to $k$-means clustering problem. In this section, we extend the HST to $k$-means and provide some brief analysis similar to $k$-median. We present the analysis in the non-private case, which can then be easily adapted to the private case. Define the following costs for $k$-means.

$$cost_{km}^T(U) = \sum_{y \in U} \min_{x \in C_0} \rho^T(x, y)^2, \tag{12}$$

$$cost_{km}^T{}'(U, C_1) = \min_{|F \cap T(v)|=1, \forall v \in C_1} \sum_{y \in U} \min_{x \in F} \rho^T(x, y)^2, \tag{13}$$

$$OPT_{km}^T(U) = \min_{F \subset U, |F|=k} \sum_{y \in U} \min_{x \in F} \rho^T(x, y)^2 \equiv \min_{C_1'} cost_{km}^T{}'(U, C_1'). \tag{14}$$

For simplicity, we will use $cost_{km}^T{}'(U)$ to denote $cost_{km}^T{}'(U, C_1)$ if everything is clear from context. Here, $OPT_{km}^T$ (14) is the cost of the global optimal solution with 2-HST metric.

**Lemma D.1** (Subtree search). $cost_{km}^T{}'(U) \leq 17OPT_{km}^T(U)$.

*Proof.* The analysis is similar with the proof of Lemma 3.3. Thus, we mainly highlight the difference. Let us just use some notations the same as in Lemma 3.3 here. Let us consider the clustering with center set be $C^* = \{c_1^*, c_2^*, ..., c_k^*\}$ (each center $c_j^*$ is a leaf of subtree whose root be $c_j'$), and $S_j'$ be the leaves assigned to $c_j^*$ in optimal k-means clustering in tree metric. Let $S_j$ denote the set of leaves in $S_j'$ whose distance to $c_j^*$ is strictly smaller than its distance to any centers in $C_1$. Let $P_j$ denote the union of paths between leaves of $S_j$ to its closest center in $C_1$. Let $v_j''$ be the nodes in $P_j$ with highest level satisfying $T(v_j'') \cap C_1 = \emptyset$. The score of $v_j''$ is $2^{h_{v_j''}} N(v_j'')$. That means the swap with a center $v_j'$ into $C_1$ can only reduce $(4 \cdot 2^{h_{v_j''}})^2 N(v_j'')$ to $cost_{km}^T{}'(U)$. We just use $v_j'$ to represent $v_j''$ for later part of this proof for simplicity. By our reasoning, summing all the swaps over $C_1^* \setminus C_1$ gives

$$cost_{km}^T{}'(U) - OPT_{km}^T(U) \leq \sum_{v_j' \in C_1^* \setminus C_1} N_{v_j'} \cdot (4 \cdot 2^{h_{v_j'}})^2,$$

$$OPT_{km}^T(U) \geq \sum_{v_i \in C_1 \setminus C_1^*} N_{v_i}(2^{h_{v_i}})^2.$$

Also, based on our discussion on Case 1, it holds that

$$N_{v_j'}2^{h_{v_j'}} - N_{v_i}2^{h_{v_i}} \leq 0.$$

Summing them together, we have $cost_{km}^T{}'(U) \leq 17OPT_{km}^T(U)$.

$\square$

Next, we show that the greedy leaf search strategy (Algorithm 3) only leads to an extra multiplicative error of 2.

**Lemma D.2** (Leaf search). $cost^T_{km}(U) \leq 2cost^T_{km}{}'(U)$.

*Proof.* Since the subtrees in $C_1$ are disjoint, it suffices to consider one subtree with root $v$. With a little abuse of notation, let $cost^{T'}_1(v, U)$ denote the optimal $k$-means cost within the point set $T(v)$ with one center in 2-HST:

$$cost^{T'}_1(v, U) = \min_{x \in T(v)} \sum_{y \in T(v)} \rho^T(x, y)^2, \tag{15}$$

which is the optimal cost within the subtree. Suppose $v$ has more than one children $u, w, ...$, otherwise the optimal center is clear. Suppose the optimal solution of $cost^{T'}_1(v, U)$ chooses a leaf node in $T(u)$, and our HST initialization algorithm picks a leaf of $T(w)$. If $u = w$, then HST chooses the optimal one where the argument holds trivially. Thus, we consider $u \neq w$. We have the following two observations:

- Since one needs to pick a leaf of $T(u)$ to minimize $cost^{T'}_1(v, U)$, we have $cost^{T'}_1(v, U) \geq \sum_{x \in ch(v), x \neq u} N_x \cdot (2^{h_x})^2$ where $ch(u)$ denotes the children nodes of $u$.

- By our greedy strategy, $cost^T_1(v, U) \leq \sum_{x \in ch(u)} N_x \cdot (2^{h_x})^2 \leq cost^{T'}_1(v, U) + N_u \cdot (2^{h_u})^2$.

Since $h_u = h_w$, we have

$$2^{h_u} \cdot (N_u - N_w) \leq 0,$$

since our algorithm picks subtree roots with highest scores. Then we have $cost^T_1(v, U) \leq cost^{T'}_1(v, U) + N_w \cdot (2^{h_w})^2 \leq 2cost^{T'}_1(v, U)$. Since the subtrees in $C_1$ are disjoint, the union of centers for $OPT^T_1(v, U)$, $v \in C_1$ forms the optimal centers with size $k$. Note that, for any data point $p \in U \setminus C_1$, the tree distance $\rho^T(p, f)$ for $\forall f$ that is a leaf node of $T(v)$, $v \in C_1$ is the same. That is, the choice of leaf in $T(v)$ as the center does not affect the $k$-median cost under 2-HST metric. Therefore, union bound over $k$ subtree costs completes the proof. $\square$

We are ready to state the error bound for our proposed HST initialization (Algorithm 2), which is a natural combination of Lemma D.1 and Lemma D.2.

**Theorem D.3** (HST initialization). $cost^T_{km}(U) \leq 34OPT^T_{km}(U)$.

We have the following result based on Lemma 3.2.

**Theorem D.4.** *In a general metric space,*

$$E[cost_{km}(U)] = O(\min\{\log n, \log \triangle\})^2 OPT_{km}(U).$$

