# OpenReview forum: "k-Median Clustering via Metric Embedding: Towards Better Initialization with Privacy"
_ICLR.cc/2022/Conference — ICLR 2022 Submitted_

### Official Review · Reviewer_o7Pb · 2021-11-02

**Correctness:** 4
**Technical Novelty And Significance:** 2
**Empirical Novelty And Significance:** 2
**Recommendation:** 5
**Confidence:** 4

**Main Review:**

The paper uses familiar techniques from metric embedding literature to suggest an algorithm for the k-median problem. The contributions does not seem to be very strong.
- The Metric embedding based algorithm for k-median is shown to give approximation guarantee of log{min(k, d)} which is better than k-means++ which gives O(log k). However, there are other algorithms that give much better approximation guarantees. It is not clear why the comparison is done with k-means++ here. This is something that the paper does not elaborate. Perhaps the algorithm is being suggested as an initialisation routine and hence the comparison is done with k-means++ but then that cannot be the only reason since the other algorithms with better approximation guarantees can also be suggested as initialisation routines. The discussion seems to be lacking on this aspect in my opinion.
- The improvement with respect to the Differentially Private seems to be minor over the previous work.

**Summary Of The Paper:**

The paper suggests an algorithm for the metric k-median problem using ideas from Metric embedding theory. The suggested use of the algorithm is as an initialization routine for the local search based algorithm for k-median. The differentially private version of the algorithm is also given along with bounds on k-median approximation factor. Experiments are conducted over datasets such as MNIST and results compared against the k-means++ algorithm (a popular initialisation algorithm).

**Summary Of The Review:**

In summary, the paper neither introduces new techniques nor obtains significant improvement over past results. My suggestion for improving the paper would be to add a discussion on why the suggested algorithm should be the right "initialisation" algorithm.

---

> ### Author Response · Authors · 2021-11-19
> **Response to Reviewer o7Pb**
>
> Dear Reviewer,
>
> We appreciate your valuable feedback.
>
> 1. Thanks for the suggestion to add more discussion on this regard. We have added the clarification in the revised paper (Section 3.1). Indeed, as you kindly pointed out, there are several tree methods that can achieve constant error in tree metric in (under non-private setting), e.g., [1,2]. However, the major bottleneck would be the computational cost (we have added the discussion and experiments on the complexity in the revision). The main motivation and advantage of our framework can be clarified from the non-private and the private setting respectively:
>
> (i) Non-private setting. There are tree methods [1,2] that can find exact optimal solution in the tree metric. However, the running time of such tree search algorithms are $O(kn^2)$ and $O(k^2nh)$ ($h$ is at least $O(\log n)$), while our algorithm only takes $O(n\log n)$. Typically, one would expect fast running speed of an initialization method (otherwise it becomes less meaningful). To this end, the $k$-median++ is the most popular initialization method for clustering, which achieves good approximation error and fast speed. Our proposed search procedure on the HST is very simple, and at the same time also reaches a constant error in the tree metric. Overall, it may achieve comparable error and speed as $k$-median++ in the non-private setting.
>
> (ii) Another important motivation of our HST method is that we can easily extend it to the private setting. To our knowledge, there has not been any initialization routine that has rigorous differential privacy guarantee.
> Our DP-HST improves previous error bound theoretically, and also performs better empirically.
>
>
> 2. Our paper makes the following contribution to the existing literature on private clustering:
>
> (i) Firstly, our Algorithm 4 and Theorem 4.2 provide a new efficient strategy based on HST without running differentially private (DP) local search, which enjoys provably small approximation error ($O(\log n)$) and rigorous differential privacy guarantee. We believe this simple algorithm itself would be interesting and useful to the design of clustering algorithms.
>
> (ii) Secondly, to our best knowledge, Theorem 4.3 is the first result to improve the error of differentially private local search derived by [3]. We achieve this by our refined DP-HST initialization approach. Notably, our additive error is fairly close to the theoretical lower bound, only by a factor of $k\log\log n$, and whether we can reach sharper errors becomes an interesting future problem.
>
> (iii) Practically, as shown by our additional experiments in Appendix B.3, the theoretical improvement of Theorem 4.3 on the iteration cost is indeed beneficial. DP-HST needs fewer iterations than $k$-median++ to find a ''local optimal'' solution. Thus, we may potentially save more computations as suggested by our theory.
>
> We hope that our rebuttal can well address your questions. Thanks again for your valuable feedback.
>
> [1] - Shah, Rahul. Faster algorithms for k-median problem on trees with smaller heights. Technical report. 2003.
>
> [2] - Tamir, Arie. An algorithm for the -median and related problems on tree graphs. Operations Research Letters. 1996.
>
> [3] - Anupam Gupta,  Katrina Ligett,  Frank McSherry,  Aaron Roth, and Kunal Talwar. Differentially private combinatorial optimization, SODA 2010.

---

> ### Author Response · Authors · 2021-11-30
> **Feedback on our response?**
>
> Dear Reviewer,
>
> Again, we appreciate your efforts in reviewing our paper. As the deadline of the discussion period is approaching, we would like to know if you have any additional feedback?
>
>  In the rebuttal and paper revision, as you kindly suggested, we added the compexity analysis and comparison with other exact tree search methods to better explain the motivation and advantage of our HST initialization method. Our algorithm achieves constant error in tree metric efficiently, and can be easily extended to differential privacy.
>
> We hope our response (and the revision) has addressed your concerns. Please let us know if there are further questions which we could help clarify. Thank you.
>
> Sincerely,
>
> Authors

---

### Official Review · Reviewer_xvPV · 2021-11-02

**Correctness:** 3
**Technical Novelty And Significance:** 3
**Empirical Novelty And Significance:** 3
**Recommendation:** 6
**Confidence:** 4

**Main Review:**

The main strengths of this paper lie in the differentially private version of their algorithm.  Here we get both theoretical and empirical improvements over prior work.  I also appreciated the simplicity of the algorithms and the clarity of the presentation.

In terms of weakness, first there is some misleading language with how the result for the standard setting is presented.  On page 2, the paper claims that the proposed method provides an $O(\log \min(k,d))$-approximation, but later on page 6 this is clarified as being an $O(\log(\min(\Delta,k)))$-approximation.  This achieves the former bound when the input data is bounded so that $\Delta = O(d)$, but this caveat is not discussed in the beginning of the paper.  I recommend the authors to move this discussion up to the statement of their contributions to avoid misleading readers.

Next, the main approach of the paper is to embed the input metric into a tree metric (via an HST) then efficiently compute an O(1)-approximation on this HST.  Metric embeddings (especially tree embeddings) are a standard technique in approximation algorithms, and this should be made clear in the discussion.  Additionally, it should be noted that there are polynomial time algorithms for computing an exact k-median solution on a tree metric (e.g. see [1,2]).  The proposed algorithm is a worthwhile contribution due to its simplicity, but these prior results should be discussed.  Additionally, it would be interesting to use one of these exact methods as a baseline in the experiments.

In the experiments, the authors run a fixed 20 iterations of local search after finding the initial centers, then report the k-median costs.  It might be interesting to also compare the runtime/iteration cost of reaching a locally optimal solution for the proposed methods and baselines, as well as the final cost of the locally optimal solutions found by each.  This sort of experiment seemed to be motivated by the discussion of the improved iteration bound for the differentially private method, but is missing.

As of now I lean more towards rejection, but would be inclined to increase my score of the above comments are addressed.

References

[1] - Shah, Rahul. Faster algorithms for k-median problem on trees with
smaller heights.  Technical report. 2003.

[2] - Tamir, Arie.  An $O(pn^2)$ algorithm for the $p$-median and related problems on tree graphs.  Operations Research Letters. 1996.

--------------------------------------------------

Edit after reading the author's responses:

One of the main benefits of the proposed method for clustering on a tree is that it can be adapted to the differentially private setting, which it is unclear how to do for other (dynamic programming based) methods.  I have raised my score to weak accept.




**Summary Of The Paper:**

This paper considers the problem of finding good initial centers for the fundamental problem of $k$-median clustering using a randomized embedding of the original metric into a tree metric.  After setting the initial centers, a standard local search algorithm is applied to produce an improved solution.  This is explored in both the standard context of $k$-median clustering, as well as in the relevant context of differentially private clustering.  In the latter setting, the goal is to minimize the amount of additive error introduced by the algorithm subject to being $\epsilon$-differentially private.  An extension to $k$-means is given in the appendix.

In the standard setting of $k$-median clustering, the main theoretical result is an initialization algorithm which is an $O(\log(\min(\Delta,k)))$-approximation to the optimal k-median clustering.  This is an improvement over k-median++ (which gives $O(\log k)$) when $\Delta$ is small, e.g. for $\Delta = O(d)$ and $d$ is small.  Using this as a seed results for a local search method results in an $O(1)$-approximation overall.  At a high level, their algorithm first constructs an embedding of the original metric into a hierarchically well-separated tree (HST).  From there, the initialization can be seen as finding an $O(1)$-approximate solution on the HST efficiently.  The overall guarantee follows from standard results about HST's

In  the differentially private setting, the main result is a similar guarantee on the quality of the initial solution and also a bound on the quality of the final solution when using a known private local search algorithm.  The quality of the final solution has $O(1)$-multiplicative error and $O(\epsilon^{-1}k^2\Delta\log(n)\log\log(n))$ additive error.  This is an improvement over the additive error of $O(\epsilon^{-1}k^2\Delta\log^2(n)$ due to Gupta et al. 2010.  The number of local search iterations is also improved from $O(k\log n)$ to $O(k\log\log n)$.  The main idea for the initialization is similar to the standard setting, but here they use the structure of the HST to ensure the initial solution is private by injecting a different amount of noise at each level of the tree.

An empirical study is done on a class of synthetic graphs as well as the MNIST dataset.  For the synthetic graphs, the metric space is given by the weighted shortest path distance in each graph, while for the MNIST dataset the metric is given by either $\ell_1$ or $\ell_2$.  The authors compare both the initial costs and the final costs (after running a local search method) for several initialization methods in both the standard and differentially private settings.  The main observation is that the proposed initialization methods tend to have better initial cost and the proposed differentially private method often outperforms the other methods in both initial cost and final cost.





**Summary Of The Review:**

This paper considers initialization methods for $k$-median clustering in both the standard setting and the differentially private setting.  The paper gives theoretical bounds for their methods in both settings and backs this up with an empirical study.  Given the misleading presentation of some of the results, a lack of discussion/comparison to prior work on k-median in tree metrics, and a lack of running time/iteration count comparison in the experiments, I am not okay with accepting this paper unless these points are addressed.

---

> ### Author Response · Authors · 2021-11-19
> **Response to Reviewer xvPV**
>
> Dear Reviewer,
>
> Thanks for your valuable feedback.
>
> 1. Thanks for the suggestion. We have modified the statement in the introduction for better clarification in the updated version.
>
> 2. Thanks for mentioning these two references on classical tree search methods for clustering. We have added the comparison to these two methods, as well as the analysis on the time complexity of our algorithm in the revision. Here is a brief explanation on the important differences and the motivation (advantage) of our method:
>
> (i) As you kindly pointed out, both dynamic programming algorithms in [1,2] have higher complexity. After the HST is built, the running time of [2] is $O(kn^2)$ quadratic in $n$, and the complexity of [1] is $O(k^2nh)$ ($h$ is height of tree, which is at least $O(\log n)$). In practice, this could fairly slow. That is why we proposed a new and efficient tree search method in service of our initialization framework. Although [1,2] achieve exact optimal $k$-median cost in the tree metric, they anyway only reduces a constant error compared to ours. Being translated into the general metric, if these two methods are used as initialization, we eventually would still get the $O(\log \min\{n,\triangle\})$ error, but with much higher computational overhead compared with our tree search algorithm with $O(n\log n)$ complexity. Typically, one would not want to spend too much time for initialization. Thus, the advantage of our method is that, we can efficiently find initial centers with good enough error (constant error in tree metric), which serves well the purpose of finding a ''roughly'' good center set to start local search.
>
> (ii) Another important motivation of our method is the differential privacy. It is unknown how to apply the exact dynamic programming algorithms in [1,2] to the private setting. Thus, it seems hard to implement them in the DP setting. If we run [1,2] on the universe set $U$, we expect them to perform similarly to $k$-median++ as in Figure 3 and Figure 4. In contrast, our DP-HST method is able to find good initial centers in the private case adapted to the problem, theoretically improving the additive error and empirically outperforming other popular initialization methods like $k$-median++.
>
> 3. Experiments on the iteration cost. Thanks for the great suggestion to add the results on the iteration cost. We agree that such numerical experiments would be useful. We have included the experiments as required in the revised version, Appendix B.3. In the private setting, since the $k$-median cost does not monotonically decrease (due to the probabilistic exponential mechanism), we compute the moving average of the costs with window size 5, and define the ''local optimum'' as the minimum of the moving average. From the Figure 6, we see that our algorithm achieves the local optima (i.e., a good solution) with fewer iterations than $k$-median++ and random initialization. In Figure 7, we also plot the cost corresponding to the optimal solution found by each algorithm as suggested. Similar to Figure 2, we see that DP-HST consistently achieves lower error than the other two methods.
>
> We hope that our explanation and additional experiments can adequately address your concerns. Thanks again for your valuable feedback.
>
> [1] - Shah, Rahul. Faster algorithms for k-median problem on trees with smaller heights. Technical report. 2003.
>
> [2] - Tamir, Arie. An algorithm for the -median and related problems on tree graphs. Operations Research Letters. 1996.

---

> > ### Comment · Reviewer_xvPV · 2021-11-23
> > **Response to response**
> >
> > Thank you for the detailed response.  I see that one benefit of the proposed k-means algorithm for HST's over exact dynamic programming on a tree is that it lends itself to the differentially private implementation better.
> >
> > In terms of the running time of the proposed method, the revision states it is $O(dn \log n)$ in the euclidean case.  What is the running time for general metrics?

---

> > > ### Author Response · Authors · 2021-11-24
> > > **Thanks for the feedback**
> > >
> > > Dear Reviewer,
> > >
> > > Thanks for your response. We are glad that our response has addressed your questions.
> > >
> > > We use the Euclidean space as an example for a convenient comparison since in the original paper of k-median++, the authors also gave the complexity in the Euclidean space. In general metric space, e.g. on a graph with $n$ nodes and $m$ edges, the complexity of k-median++ is roughly $O(km+kn\log n)$ as we need to compute the shortest path distances of $k$ centers to other points. An HST can be constructed in $O(m\log n)$ time, see [1]. The time cost for the tree search algorithm is $O(n\log n)$, so the total cost of our initialization method would be $O(m\log n+n\log n)$. We see that HST initialization is always faster when $k\geq \log n$, same as in the Euclidean case. Moreover, it would also be more efficient when $k<\log n$ but satisfies $(\log n-k)m<(k-1)n\log n$.
> > >
> > > We would be happy to add this complexity comparison in the general metric space to the paper. Please let us know if there are more questions.
> > >
> > > Thanks again for your feedback and suggestions.
> > >
> > >
> > > [1] - Guy E. Blelloch, Yan Gu, Yihan Sun, Efficient Construction of Probabilistic Tree Embeddings. ICALP 2017.

---

### Official Review · Reviewer_3Abh · 2021-11-03

**Correctness:** 4
**Technical Novelty And Significance:** 3
**Empirical Novelty And Significance:** 2
**Recommendation:** 6
**Confidence:** 3

**Main Review:**

Strengths:
- This paper gives improved k-medians bounds in the general metric setting (non DP) that improve over the literature in the d < k regime. It also gives the best known theoretical guarantee in the DP setting, and is worse than the lower bound by a small additive factor (k log log n)
- The paper empirically compares their algorithm to a number of standard baselines, showing favorable results for multiple datasets and metrics, and especially for less separable data and when the input data an unbalanced subset of the universe.

Weaknesses:
A few comments on improving the paper:
- The paper does not discuss runtimes of algorithms. A discussion of runtimes as well as comparison of runtimes in experiments would be useful
- Regarding the results for k-means - moving the k-means results to the main paper, as well as a brief discussion and comparison to k-medians would be useful.

We give a number of suggestions to improve clarity:
- “HST tree” -> “HST”
- “symmetric difference one” -> “symmetric difference of size one”
- (In Algorithm 1) “cost(F - x + y)” -> “cost(F - {x} + {y})”
- “we will count levels from large to small” -> “we will count levels in descending order down the tree”
- (Section 3.2 Intro) - “Suppose T is an L = log \Delta -level-2 HST” -> “Let L = log \Delta and suppose T is an L-level-2-HST”
- (Algorithm 2) Mention Algorithm 6 in “Build a level-L 2-HST tree T based on input U”
- State Theorem 3.4 and Theorem 3.5 before lemmas
- State that NDP stands for Non Differentially Private
- Theorem 4.2: Constants (10) can be absorbed into big-oh notation


**Summary Of The Paper:**

This paper introduces a new initialization scheme for the k-medians clustering problem in the general metric space setting. This is based on the construction of metric embeddings via 2-HST’s (Hierarchically well-separated trees). The authors also extend this to the differential privacy (DP) setting. They prove approximation guarantees in both the non-DP and DP settings, improving upon the literature. Finally, they empirically validate algorithms against a number of baselines with both real world and synthetic datasets for multiple metrics.

**Summary Of The Review:**

This paper improves upon the literature for the k-medians clustering problem in the general metric setting as well as in the differentially private case. The approximation guarantees improve the best known results in this case. The experiments demonstrate the validity of the theoretical contributions as well.

---

> ### Author Response · Authors · 2021-11-19
> **Response to Reviewer 3Abh**
>
> Dear Reviewer,
>
> We sincerely appreciate your valuable feedback and suggestions. We have corrected the typos in the revision.
>
> 1. Running time: Thanks for the great suggestion. We have added the analysis (Proposition 3.1) and numerical comparison (Appendix B.2) on the complexity of HST and $k$-median++ in the revised paper.
>
> It is known that constructing a 2-HST needs $O(d n \log n)$ time ([Bartal, 1996]), in the Euclidean space for example. If we use a priority queue to store $C_1$, the subtree search step of our HST initialization algorithm involves at most sorting all the nodes in HST based on score, which takes $O( n\log n)$. Hence, the complexity of this step is $O(d n\log n)$. The leaf search step to obtain $C_0$ only takes at most $O(n)$. Thus, the total time of HST initialization is $O(d n\log n)$. On the other hand, the standard implementation of $k$-median++ has time complexity $O(d n k)$ [Arthur and Vassilvitskii, 2007]. Therefore, the complexity of our HST initialization is in general comparable to $k$-median++ ($\log n$ v.s. $k$). Our algorithm would be faster if $k$ is larger.
>
> 2. Thanks for the suggestion. Since (i) we hope to present the detailed algorithm design and the path to our theoretical results in the current submission, and (ii) we can directly compare our results with [Gupta et al., 2010], we presented our method in the $k$-median setting and placed the extension to $k$-means in the appendix due to the limited space. The analytical tools are similar. We have added a brief comparison and discussion in the revised paper.
>
> We hope our rebuttal and paper revision can well answer your questions. Thanks again for your valuable comments.
>
> [Bartal, 1996] - Yair Bartal. Probabilistic approximations of metric spaces and its algorithmic applications, FOCS 1996.
>
> [Arthur and Vassilvitskii, 2007] - Arthur, D., Vassilvitskii, S. k-means++: the advantages of careful seeding, SODA 2007.
>
> [Gupta et al., 2010] - Anupam Gupta,  Katrina Ligett,  Frank McSherry,  Aaron Roth, and Kunal Talwar. Differentially private combinatorial optimization, SODA 2010.

---

> > ### Comment · Reviewer_3Abh · 2021-12-02
> > **Response to Author comments**
> >
> > Thank you for your response and the details on the runtime and analysis.

---

### Official Review · Reviewer_dRWX · 2021-11-07

**Correctness:** 4
**Technical Novelty And Significance:** 3
**Empirical Novelty And Significance:** 3
**Recommendation:** 6
**Confidence:** 3

**Main Review:**

Overall the paper makes some good contributions and adds to the communities' understanding of the k-clustering problem, espeically with privacy constraints. The paper proposed new algorithm designs and proved updated bounds for k-median with/without privacy constraints.  The experiment design also seems comprehensive and persuasive to me.

The writing in this paper is mostly smooth but could still be improved and be more clarifying in some ways. The paper did well in presenting the algorithm's framework and the ideas there are interesting. Although one can argue that the improvement in approximation ratio in classical k-median clustering is marginal, but as the authors noted, the design is new and it serves well for the setting of differential privacy constraints. I do think the authors could be more clear when talking about previous work on privacy clustering, the approaches used and the differences, the setting of Euclidean space v.s. general graph input. Right now the problem setting and the results seem a bit confusing to me.

Minor comments:
1. Does the number "2" really matter in the 2-HST that you are using, or can we replace it with something like $1+\epsilon$?
2. In the approximation ratio there is $min\{k, d\}$. Does that mean that we can only obtain this result with points in Euclidean space?
3. The notation $N(v)$ and $N_v$ share the same meaning, right?
3. When defining $score(v)=N(v)\cdot 2^{h_v}$, is this the first time the notation $h_v$ appears? I couldn't seem to find a definition of it.
4.  In my opinion the paper sometimes uses a new terminology and assume the reader knows what it is. I think it is better to introduce terminologies and include a short description of them just to make sure the reader is on the same page. For example, the major comparison, k-means/median++,  is never fully explained.
5. The paper studies k-median clustering, I wonder if we know anything about what happens when we switch to k-means. Do the conclusions still hold?

**Summary Of The Paper:**

The paper proposes a new initialization scheme for the k-median problem on graph input (or general metric spaces) using metric embedding tree structure. The paper proposes an algorithm that finds initialization of good centers using HST that gets an approximation factor of O(log min{k,d}) if the data is in Euclidean space  where d is the number of dimensions. Then, the paper studies clustering with differential privacy guarantee and hows that the initialization method could be adapted to give a slightly stronger muliplicative and additive errors. The work complemented these theoretical findings with experiments and show that the proposed initialization imporves the performance of k-median++ initialization.

**Summary Of The Review:**

Mostly, I consider the contribution made in this paper to be meaningful to the clustering community, but it could be improved (at least in writing). It has a valid theoretical framework, but as a review from the broader clustering community, I find it hard to judge how significant these findings are.

---

> ### Author Response · Authors · 2021-11-19
> **Response to Reviewer dRWX**
>
> Dear Reviewer,
>
> Thanks for your valuable comments and suggestions.
>
> Firstly, we would like to explain more on the general problem setting and significance of our work. Our contribution is to propose a novel and efficient initialization framework, based on metric embedding trees, for general metrics that can in particular work in the private setting. We would like to add several clarifications regarding your questions:
>
> (i) Distinction between ''Euclidean'' and ''general'': As we introduced in Section 1 (''Clustering with Differential Privacy''), while there are many works on private clustering, many of them consider the Euclidean spaces (with $l_p$ norms). That said, these methods either are built upon Lloyd-Max type algorithms which require computations in the Euclidean space, or use projections, parametrizations or space division that are only applicable to Euclidean spaces. Our HST initialization method works for general metrics, allowing broader applications such as clustering on graphs. In the general metric spaces, the aforementioned operations in Euclidean space can no longer be applied. For a concrete practical graph clustering problem of our setting, please see Example 2.1.
>
> (ii) Our paper is the first initialization routine (that runs efficiently, please see our revised paper for a comparison) with differential privacy in literature, which leads to a direct improvement in the approximation error of [1]. Our experiments show that both with and without privacy, HST can find better initial centers (with smaller error) than the most popular $k$-median++ initialization method.
>
> Next, we answer your specific questions as follows.
>
> 1. The ``2'' in 2-HST can be replaced by any constant larger than one, which only affects the constants in the errors but does not change the order. In the paper, we use 2-HST for notation convenience.
>
> 2. The statement in Remark 3.1 is specific to the Euclidean space (in general, spaces with bounded doubling dimension). We have clarified this in the revision. Theorem 3.5 is the result in the general metric space, where the approximation error is $\min\{\log n,\log\triangle \}$.
>
> 3. Yes, we have made it consistent in the revision. Thanks for pointing this out.
>
> 4. $h_v$ is the level of a node $v$, which is defined in the sentence on the bottom of page 3 in the original submission (top of page 4 in the updated paper), ''We use $h_v$ to denote the level of $v$, ...''.
>
> 5. Thanks for this suggestion. We have included it in the Appendix A of the revised paper.
>
> 6. Yes, the algorithm and analysis can be easily extended to $k$-means. We provided some related results in Appendix C of the original submission (Appendix D of the revised version). Basically, the approximation error in the standard non-private setting is $\log^2 n$ instead of $\log n$ of $k$-median. Similar analysis can also be applied to the DP setting.
>
> We hope our rebuttal and paper revision well answer your questions. Thanks again for your valuable comments.
>
> [1] - Anupam Gupta,  Katrina Ligett,  Frank McSherry,  Aaron Roth, and Kunal Talwar. Differentially private combinatorial optimization, SODA 2010.

---

### Author Response · Authors · 2021-11-19
**General response and the paper revision**

Dear Reviewers and Area Chairs,

We sincerely appreciate your effort in reviewing our submission and all the constructive feedback. Based on the raised questions and suggestions, we have updated the paper (please kindly check the uploaded pdf file). Besides improving the presentation and some clarifications, the main updates are summarized as follows:

(i) We updated the introduction part for better clarity, and highlighted the contribution/significance of our basic HST search algorithm (Algorithm 2). Reaching a constant error on trees as well as being efficient, we believe the search method itself would be interesting and useful to the design of clustering algorithms.

(ii) In Section 3.2, we add the complexity of our HST initialization algorithm, and the comparison with $k$-median++. In general, they have comparable speed.

(iii) In Section 3.1, we add the discussion of several related methods on exact clustering on metric trees. Those methods have high complexity that makes them unsuitable for the purpose of fast initialization, and it is unknown how to apply them to the private case effectively. Our proposed HST search method achieves $O(1)$ approximation error in tree metric while being much faster, and can be used in the private manner with good performance.

(iv) We add Appendix B.2, where we provide numerical results on the efficiency comparison of HST v.s. $k$-median++. Appendix B.3 presents the improved empirical iteration cost of DP-HST to justify Theorem 4.3.

Again, we sincerely thank you for all your valuable suggestions to help us improve the quality of the submission. We hope that our updated paper can well address your concerns.

---

> ### Author Response · Authors · 2021-11-22
> **Any additional coments?**
>
> Dear Reviewers and Area Chairs,
>
> Thanks again for the careful reviews and thoughtful comments. We have responded to the reviewers and uploaded the revised version which included 1) complexity analysis and comparison with more methods; 2) additional experiments on efficiency and improved iteration cost.
>
> Please let us know if there are additional comments which we should need to further address, in particular before the deadline (today 11/22, AOE) for further revising the paper. One of the main authors will be taking  an international flight tonight  but we will try our best to address any new issues raised before boarding.  Thank you.
>
> Best Regards,
>
> Authors

---

### Decision · Program_Chairs · 2022-01-20

**Decision:**

Reject

**Comment:**

This paper considers initialization methods for the k-means algorithm.  There is a lot of prior work in this area. The reviewers were mildly positive on the paper.  There were several concerns on how the results were presented as well as the comparison to prior work. Importantly, no reviewer felt that there was a lot of novelty in the paper over the line of work on k-means initialization.